# Highly specific and sensitive detection of *Burkholderia pseudomallei* genomic DNA by CRISPR-Cas12a

**Somsakul Pop Wongpalee**[1]*, **Hathairat Thananchai**[1], **Claire Chewapreecha**[2,3], **Henrik B. Roslund**[1], **Chalita Chomkatekaew**[2], **Warunya Tananupak**[1], **Phumrapee Boonklang**[2], **Sukritpong Pakdeerat**[2], **Rathanin Seng**[4], **Narisara Chantratita**[4], **Piyawan Takarn**[1], **Phadungkiat Khamnoi**[5]

**1** Department of Microbiology, Faculty of Medicine, Chiang Mai University, Chiang Mai, Thailand, **2** Mahidol Oxford Tropical Medicine Research Unit (MORU), Faculty of Tropical Medicine, Mahidol University, Bangkok, Thailand, **3** Parasites and Microbes Programme, Wellcome Sanger Institute, Hinxton, United Kingdom, **4** Department of Microbiology and Immunology, Faculty of Tropical Medicine, Mahidol University, Bangkok, Thailand, **5** Microbiology Unit, Diagnostic Laboratory, Maharaj Nakorn Chiang Mai Hospital, Chiang Mai, Thailand

* Pop.Wongpalee@gmail.com

**Data Availability Statement:** All relevant data are within the manuscript and its Supporting Information files.

## Abstract

Detection of *Burkholderia pseudomallei*, a causative bacterium for melioidosis, remains a challenging undertaking due to long assay time, laboratory requirements, and the lack of specificity and sensitivity of many current assays. In this study, we are presenting a novel method that circumvents those issues by utilizing CRISPR-Cas12a coupled with isothermal amplification to identify *B. pseudomallei* DNA from clinical isolates. Through *in silico* search for conserved CRISPR-Cas12a target sites, we engineered the CRISPR-Cas12a to contain a highly specific spacer to *B. pseudomallei*, named crBP34. The crBP34-based detection assay can detect as few as 40 copies of *B. pseudomallei* genomic DNA while discriminating against other tested common pathogens. When coupled with a lateral flow dipstick, the assay readout can be simply performed without the loss of sensitivity and does not require expensive equipment. This crBP34-based detection assay provides high sensitivity, specificity and simple detection method for *B. pseudomallei* DNA. Direct use of this assay on clinical samples may require further optimization as these samples are complexed with high level of human DNA.

## Author summary

Melioidosis is a fatal infectious disease caused by a Gram-negative bacterium called *Burkholderia pseudomallei*. The bacteria can be found in many parts of the world, especially in the tropical and subtropical regions. Infection displays a variety of symptoms such as pneumonia, organ abscess and septicemia. The latter can lead to death within 24–48 hours if not properly diagnosed and treated. Rapid and accurate diagnosis, consequently, are essential for saving patients' lives. Currently, culturing *B. pseudomallei* is a gold

**Funding:** This research was supported by TSRI Fundamental Fund 2021, Chiang Mai University (grant number 64A104000067 to SPW). CCho was funded by Wellcome International Master Fellowship (221418/Z/20/Z). CChe was funded by Wellcome International Intermediate Fellowship (216457/Z/19/Z) and Sanger International Fellowship. The funders had no role in study design, data collection and analysis, decision to publish, or preparation of the manuscript.

**Competing interests:** The authors have declared that no competing interests exist.

standard diagnostic method, but the assay turnaround time is 2–4 days, and the result could be of low sensitivity. Other detection methods such as real-time PCR and serological assays are limited by availability of equipment and by low specificity in endemic areas, respectively. For these reasons, in this study we developed a specific, sensitive and rapid detection assay for *B. pseudomallei* DNA, that is based on CRISPR-Cas12a system. The CRISPR-Cas12a is a protein-RNA complex that recognizes DNA. The RNA can be reprogramed to guide the detection of any DNA of interest, which in our case *B. pseudomallei* genomic DNA. Our data showed that this assay exhibited a 100% specificity to *B. pseudomallei* while discriminating against 10 other pathogens and human. The assay can detect *B. pseudomallei* DNA in less than one hour and does not require sophisticated equipment.

## Introduction

*Burkholderia pseudo*mallei is an environmental Gram-negative bacillus found mostly in soil and water, and a causative agent of melioidosis—a rapidly fatal infectious disease endemic in many tropical and subtropical countries [1–3]. Septicemic melioidosis can lead to death in less than 48 hours if not diagnosed and treated properly [4,5]. Human hosts can acquire *B. pseudomallei* by skin inoculation, ingestion of contaminated food or water supply, or inhalation of aerosolized bacterium. Infected individuals can display a variety of clinical manifestations ranging from pneumonia, organ abscesses to septicemia [3,6], which make clinical diagnosis challenging. The mortality rate of infected patients is high, reported as 35–40% in Thailand, despite availability of antibiotic treatments [1,7]. This is, in part, due to delayed- or missed diagnosis, which often leads to delayed correct treatments. The gold standard laboratory diagnosis for melioidosis remains the culturing of *B. pseudomallei* from clinical specimens, followed by a panel of biochemical tests [8,9]. This diagnosis can take several days, requires an enhanced biosafety level 2 (BSL2 enhanced) laboratory and sometimes generates inconsistent results [6,10]. Since timely and optimal treatments of melioidosis affect outcomes of the disease [3,6,10,11], rapid and accurate identification of *B. pseudomallei* is much needed.

Serological diagnosis of *B. pseudomallei*-specific antigens or antibodies, such as indirect hemagglutination (IHA), enzyme linked immunosorbent assay (ELISA), immunochromatographic assay (ICT) and lateral flow immunoassay (LFI) have been developed [12–19]. However, interpretation of these serological tests have proven to be complicated in endemic areas where past infection or prior exposure to the bacteria causes seroconversion without active infection [12,14,16,20]. In addition, heterogeneity in antibody response to *B. pseudomallei* was reported in patients, thereby requiring testing of multiple antigens to confirm the result [17]. Moreover, some *B. pseudomallei* culture-confirmed patients were seronegative during acute infection [12]. Together, these limit the application of serological assays as diagnostic tools for melioidosis.

In addition to serological assays, real-time polymerase chain reaction (real-time PCR)-based assays have been developed to detect the DNA of *B. pseudomallei* [21–25]. Although these methods showed high sensitivity and specificity, real-time PCR-based assays require expensive real-time PCR machines, which may not be readily available in many laboratories. As alternatives, isothermal DNA amplifications such as loop-mediated isothermal amplification (LAMP) [26,27] and recombinase polymerase amplification (RPA) [28–30] have been developed to overcome this dependency on expensive equipment. Both assays exhibit very sensitive detection. RPA, for example, has a limit of detection (LOD) from between 25–50 copies of *B. pseudomallei* genomic DNA. Albeit promising, these assays are based on primers

designed to target a genomic locus, such as the type 3 secretion system-1 (T3SS-1), of few selected reference strains. Their detection coverage on broader *B. pseudomallei* population is currently unknown. Furthermore, specificity of RPA assay remains to be further determined as it can tolerate some mismatches in primer sequences [31,32].

Cluster regularly interspaced short palindromic repeat (CRISPR) is an adaptive defense system in bacteria and archaea that provides sequence-specific immunity against invading nucleic acids such as bacteriophages or plasmids [33,34]. A CRISPR locus incorporates remnants of genetic material from past infections and uses them for RNA-guided endonuclease against future infections. This RNA-guided endonuclease, known as a CRISPR-Cas complex, consists of a CRISPR RNA (crRNA) and CRISPR-associated (Cas) nuclease(s). The complex recognizes a target site through crRNA-mediated base-pairing complementarity before initiating its cleavage. Various CRISPR-Cas systems have been identified, each having a unique property toward nucleic acid [35,36]. One prominent system is CRISPR-Cas9, a system that has been exploited for precise genome editing both in research fields and in gene therapy [37–39], and for potential therapeutics for infectious diseases such as human immunodeficiency virus (HIV) and hepatitis B virus (HBV) [40–42]. CRISPR-Cas12a and -Cas13a are among recently discovered members of the CRISPR-Cas family, which recognize DNA and RNA targets, respectively. Their initial target recognitions activate their non-specific endonucleases that collaterally cut other non-target nucleic acids in their vicinity [43,44]. This collateral cleavage property of the two CRISPR-Cas systems has been used as a biosensor to induce cleavage of fluorescent nucleic acid probes, allowing a sensitive and a quantifiable readout of the actual target nucleic acids [44,45]. By coupling this property of CRISPR-Cas12a or -13a with another signal enhancement step such as metal-enhanced fluorescence (MEF), autocatalysis-driven feedback amplification or an isothermal pre-amplification of nucleic acids, the CRISPR-Cas systems can be used as detection platforms that offer superior sensitivity and specificity [41,46–49]. Examples are detection assays called DNA endonuclease-targeted CRISPR *trans* reporter (DETECTR) and specific high-sensitivity enzymatic reporter unlocking (SHERLOCK), which couple isothermal pre-amplification of nucleic acids with the CRISPR-Cas12a and -Cas13a systems, respectively [44,45]. In addition, adapting a lateral flow dipstick as an assay readout further simplifies these detections to be performed rapidly in a regular laboratory setup. Consequently, both CRISPR-Cas systems have gained momentum from their applications to diagnoses of infectious diseases and biomarkers such as microRNA, circulating tumor DNA (ctDNA) and single nucleotide polymorphism (SNP) [48,50–52]. This CRISPR-Cas-based diagnostic technology has been successfully applied in the detection of various pathogens including Zika virus, Dengue virus, human papillomavirus (HPV), *Staphylococcus aureus*, *Pseudomonas aeruginosa* and most recently SARS-CoV-2, at attomolar concentrations [44–47,51,53,54].

In this study we developed a CRISPR-Cas12a diagnostic technology for *B. pseudomallei* genomic DNA. We used bioinformatic analyses to guide the design of the target site of CRISPR-Cas12a to ensure extensive coverage of the diverse *B. pseudomallei* population and to reduce cross reactions with other pathogens. We retrieved eight potential target sites of CRISPR-Cas12a from the analyses, of which the crBP34 site was programmed into CRISPR-Cas12a for further characterization. When coupling crBP34-targeting CRISPR-Cas12a with RPA in the DETECTR assay, we found that our detection platform could detect as low as 40 copies of the input *B. pseudomallei* genomic DNA. This limit of detection is comparable to the sensitivity of real-time PCR. The crBP34 target site exhibited 100% specificity when tested with DNA of *B. pseudomallei* and non-*B. pseudomallei* clinical isolates. With high specificity, sensitivity and short assay time to detect *B. pseudomallei* DNA, this detection platform is a

potential diagnostic tool for melioidosis, that warrants future developments for its application to clinical specimens.

## Methods

### Expression and purification of MBP-LbCas12a

pMBP-LbCas12a (from *Lachnospiraceae bacterium* species) was a gift from Jennifer Doudna (Addgene, USA, #113431). The plasmid was transformed into *E. coli* Rosetta 2 (DE3) (Novagen, USA, #70954). 500-mL of Luria-Bertani (LB) culture was grown at 37°C until O.D.600 reached 0.4–0.6. MBP-LbCas12a expression was induced by growing the culture in the presence of 0.1 mM IPTG overnight at 16–18°C. Bacterial cells were harvested by centrifugation, washed with PBS and resuspended in 40 mL a lysis buffer (1000 mM NaCl, 20 mM imidazole, 20 mM Tris-HCl, 5% glycerol, 1 mM DTT and 1 mM PMSF). The cells were sonicated in an ice bath using Vibra-Cell VCX 500 sonicator (Sonics & Materials, USA) at a frequency 20 kHz with 60% amplitude for 3 minutes, using 1-second pulse, 7-second rest. Cell lysate was cleared by centrifugation at 20,000 g at 4°C for 1 hour and filtered through a 0.22 um PES membrane. All protein purification steps were performed using the AKTA Pure 25 M1 FPLC system (GE Healthcare, Sweden). First, the protein was affinity-purified on a 5-mL HisTrap HP column (GE Healthcare, Sweden, #17524802) with a His binding buffer (20 mM Tris-HCl pH 7.5, 1000 mM NaCl, 5% glycerol, 1 mM DTT, 20 mM imidazole), washed with a washing buffer (20 mM Tris-HCl pH 7.5, 1000 mM NaCl, 5% glycerol, 1 mM DTT, 50 mM imidazole) and eluted with an elution buffer (20 mM Tris-HCl pH 7.5, 1000 mM NaCl, 5% glycerol, 1 mM DTT, 300 mM imidazole). Peak fractions were collected and dialyzed overnight in a dialysis buffer (20 mM Tris-HCl pH 7.5, 125 mM NaCl, 5% glycerol, 1 mM DTT, 1 mM PMSF). Soluble protein was recovered and loaded into a 5-mL HiTrap SP HP cation exchange column (GE Healthcare, Sweden, #17115201), pre-equilibrated with a binding buffer (20 mM Tris-HCl pH 7.5, 125 mM NaCl, 5% glycerol). The column was washed with the binding buffer, and protein was eluted with an ion exchange buffer (20 mM Tris-HCl pH 7.5, 5% glycerol, 125–2000 mM NaCl gradient). Fractions containing MBP-LbCas12a were combined, concentrated and injected into a HiLoad 16/600 Superdex 200 pg size exclusion column (GE Healthcare, Sweden, #28989335), pre-equilibrated with a gel filtration buffer (50 mM Tris-HCl pH 7.5, 500 mM NaCl, 5% glycerol). Protein was eluted with the same buffer. Fractions containing MBP-LbCas12a were concentrated to ~500 μl. Glycerol and DTT were added to achieve final concentrations of 20% v/v and 2 mM, respectively. The protein was aliquoted, snap frozen in liquid nitrogen and stored at -80°C.

### Searching for optimal CRISPR-Cas12a target sites

All command codes used were open-source at GitHub (https://github.com/henrikroslund/crispr-cas12a/releases/tag/BP_V2). In the first step, CRISPR selection, one of 30 representative *B. pseudomallei* genome (S2 Table), namely bb2, was used to generate an initial pool of all possible LbCas12a target sites by scanning for **TTTV**$N_1$….$N_{20}$ (PAM is indicated in bold; V is non-T nucleotide, and $N_1$ to $N_{20}$ is spacer sequence of any nucleotides). Duplicate target sites or target sites with spacer regions that contained quadruple nucleotides or GC content outside 40–65% were removed. This resulted in 26,661 candidate target sites, which were subsequently aligned to the other 29 representative *B. pseudomallei* genomes. 15,448 candidate target sites were found to be 100% conserved in all 30 genomes and were processed to the CRISPR filtering step to remove potential cross reactions with other pathogens (S5 Table). First, a pool of CRISPR-Cas12a target sites **TTTN**$N_1$….$N_{20}$ from 1,071 complete genomes of non-*B. pseudomallei* pathogens was generated. The 4th position of PAM was relaxed, so that weaker cross-

reactive target sites were generated [55]. The candidate target sites from *B. pseudomallei* were aligned to target sites from non-*B. pseudomallei* pathogens using only $N_1$ to $N_{20}$, and candidates with 100% match were removed. This initial filtering step significantly trimmed down the number of *B. pseudomallei* candidate target sites to 1,982. The resulting candidates were further filtered out by re-mapping to the above pool of target sites from non-*B. pseudomallei* pathogens. Mappability was defined as having at least 15 nucleotides matching in $N_1$ to $N_{20}$. Mapped candidates were analyzed for mismatches relative to their cross-reactive target sites. Candidates that failed to fulfill the following 'mismatch criteria' at any mapped sites were immediately removed: (i) 2 consecutive in Seed ($N_1$ to $N_6$), or (ii) 3 or more in Seed, or (iii) 4 or more in $N_7$ to $N_{20}$. The remaining candidates and candidates that were unable to be mapped to any cross-reactive target sites were considered as the final candidates (Fig 1D). Spacer regions (excluding PAM) were later engineered into crRNA.

## crRNA synthesis

DNA templates were synthesized and desalted by IDT DNA technologies (Singapore). All crRNAs were *in vitro* transcribed from annealed oligo templates using a HiScribe T7 High Yield RNA Synthesis Kit (NEB, USA, #E2040S), in accordance with the manufacturer's protocol. DNAse I (RNase-free) (NEB, USA, #M0303S) was added to the reaction, and the tube was incubated at 37°C for 20 minutes to enable the degradation of the DNA template. crRNAs were purified in 12% v/v urea-PAGE (29:1) (in 7.5 M urea, 0.5x TBE), eluted in a gel elution buffer (300 mM sodium acetate pH 5.2, 1 mM EDTA pH 8.0, 0.1% v/v SDS) and precipitated in ethanol. crRNA was resuspended in a folding buffer (10 mM Tris-HCl pH 7.5, 50 mM NaCl), folded by heating in a metal heat block at 90°C for 2 minutes and allowed to slowly cool down at room temperature. crRNA was aliquoted and stored at -80°C until used.

## Identification of representatives for major *B. pseudomallei* lineages

A global collection of *B. pseudomallei* (n = 3,341) was compiled from the public database [56–94] and other short-read sequenced data (S1 Table). When the assemblies were not available, short reads were de *novo* assembled using Velvet [95], giving a median of 136 contigs (range 60–434), and an average total length of 7,140,818 bps (min = 6,989,389 bps and max = 7,441,976 bps). Gene predictions and annotations of assemblies were performed using Prokka [96]. With an average of 5,850 (range 5685–6275 per each genome) predicted coding sequences (CDS) assigned, the assemblies fall in a similar range of 6,332 CDS and length of 7,247,547 bps as reported in the reference genome K96243 [71]. Population structure was defined using three independent approaches–PopPUNK [97], a phylogenetic tree, and multi-dimensional scaling as described previously [61]. All three methods showed high consistency and resolved the *B. pseudomallei* population into 324 lineages, many of which are singletons or lineages with a small number of isolates. We noted that there are 22 major lineages dominating the global *B. pseudomallei* population (2,429/3,341 isolates representing 73% of the dataset). To ensure that the CRISPR-Cas targets cover the genetic diversity of the entire population, candidate genomes were selected to represent each major lineage (*Burkholderia* Bayesian cluster (bb) 1 to 22 (Fig 1B), except for bb3 and bb4 which shared close ancestry with bb5 where one representative was selected to represent three lineages. We also randomly selected candidate genomes outside the major lineages. Collectively, a total of 30 genomes were selected. On average, each candidate could be assembled into 92 contigs, thereby enabling a fair length of conserved regions for the searching of CRISPR-Cas targets. The PopPUNK phylogenetic tree (Fig 1B) was visualized and annotated with the metadata using Tree of Life Tools [98].

### *In silico* detection of CRISPR-Cas targets across the genomes of *B. pseudomallei* population, the human reference genome and the representative bacterial genomes from NCBI database

An *in silico* validation of CRISPR-Cas targets were performed by mapping the designed RPA primers and target sequences for CRISPR-Cas12a against the assemblies of 3,341 *B. pseudomallei* genomes (S1 Table) using BLAT v. 36. To enable mapping of sequences with low complexity, the command "blat -minScore = 10 -tileSize = 8 <assembly> <candidate target> <out.psl>" was employed. As CRISPR-Cas12a recognition seems to be flexible at the 4th position of PAM [55] and seem to allow one sequence mismatch in the position N1 to N20 (though the activity is reduced) [99,100], we further conditioned the BLAT results to allow for two types of mismatch. For the sequence target $TTTVN_1N_2N_3....N_{20}$, we allowed for (i) a mismatch at position V; or (ii) a mismatch at any of the position from $N_1$ to $N_{20}$; or a combination of (i) and (ii).

To test for possible cross-reactivity of these CRISPR-Cas12a target sequences with human host DNA or other bacterial species, we additionally mapped them against a human reference genome GRCh38 [101] and 40,827 bacterial reference genomes archived in NCBI repository [102] (date retrieved 5th May, 2022). To reduce computational load, the search was performed using NCBI web BLASTN with results filtered to allow for (i) a mismatch at position V; or (ii) a mismatch at any of the position from $N_1$ to $N_{20}$; or a combination of both as outlined earlier. None of the hits passed the filtering threshold and are unlikely to be recognized by CRISPR-Cas12a, given that a single mismatch in $N_1$ to $N_{20}$ already significantly reduces the enzyme's function [99,100]. The analysis was summarized as S3 and S4 Figs.

### Recombinase Polymerase Amplification

RPA reagent was purchased from TwistDx (USA). All RPA primers (S3 Table) were 35-nt long with amplicon length less than 500 base-pairs (ideally 100–200 base-pairs), as recommended by TwistDx. These primers were manually designed and checked for their secondary structures and homo-/hetero- duplex using RNAfold (http://rna.tbi.univie.ac.at/cgi-bin/RNAWebSuite/RNAfold.cgi) and IDT OligoAnalyzer (https://www.idtdna.com/pages/tools/oligoanalyzer), respectively. Primers were synthesized and purified by standard desalting by Macrogen (Korea). TwistAmp Basic (TwistDx, USA, #TABAS03KIT) and TwistAmp Liquid Basic (TwistDx, USA, #TALQBAS01) were both used in this study in accordance with the manufacturer's protocols, with the following modifications: (i) total volume of an RPA reaction was adjusted to 30–50 μl and (ii) incubation was done at 39˚C for 30 minutes. For the specificity test (Fig 2B, 2C and 2D), ~10,000 copies of bacterial genomic DNA were used in each reaction. The copy number was approximated from the median genomic size of each bacterial species. For the sensitivity test (Fig 3A and 3B), the accurate copy number of *B. psuedomallei* (isolate 13D) genomic DNA stock was determined by real-time PCR against purified standards. The genomic DNA was then diluted to appropriate concentrations and immediately used in the RPA reaction. For detection of clinical isolates (Fig 4A), 1 ng of DNA was used per RPA reaction. All RPA reactions were stored at -20˚C until used.

### DETECTR assay with fluorescence readout

CRISPR reactions were performed in a final 100-μl volume that contained 100 nM crRNA, 200 nM MBP-LbCas12a, 500 nM FAM-Quencher probe and 1x CR buffer 1 (10 Tris-HCl pH 8.0, 50 mM NaCl, 13 mM MgCl2, 1% v/v glycerol). First, CRISPR-Cas12a ribonucleoprotein (RNP) was pre-assembled in 30-μl volume with crRNA, MBP-LbCas12a and CR buffer 1. The

reaction was incubated at room temperature for 15 minutes. A 70-µl mixture of FAM-Quencher probe with various amounts of RPA (see figure legends) was added to the RNP. This CRISPR reaction was transferred to a fluorescence plate reader Synergy H4 (BioTek, USA) and read at 483/530 nm (excitation/emission) every 3 minutes for 3 hours at 37˚C.

### DETECTR assay with lateral flow dipstick readout

RNP complex assembly and CRISPR reactions were combined and performed in a final 50-µl volume that contained 1x CR buffer 1, 100 nM crRNA, 200 nM MBP-LbCas12a, 100 nM FAM-Biotin probe and 5 µl RPA. The reaction was incubated at 37˚C for 20 minutes. A Hybri-Detect lateral flow dipstick (Milenia Biotek, Germany, #MGHD1) was directly dipped into the reaction and allowed to develop for 2–5 minutes.

### Real-time PCR

Real-time PCR primer (S3 Table) were reported previously [23]. The primers were synthesized and desalted by Macrogen (South Korea). 20-µl real-time PCR reactions were prepared using Maxima SYBR Green/ROX qPCR Master Mix (ThermoFisher Scientific, USA, #K0221) in accordance with the manufacturer's instruction. Real-time PCR analysis was performed on the 7500 Fast Real-Time PCR System (Applied Biosystems) with the following conditions: 95˚C (10 minutes); 40 cycles of 95˚C (15 seconds), 61˚C (30 seconds), 72˚C (30 seconds). Standard melting curve analysis was also performed at the end of the PCR. ΔRn threshold was automatically calculated by the built-in program.

### Bacterial isolates, culture and genomic DNA extraction

Non-*B. pseudomallei* clinical isolates were obtained from Diagnostic Laboratory, Maharaj Nakorn Chiang Mai hospital. Colonies were directly scrapped off sheep-blood agar plates and resuspended in 200 µl water. A GeneJet genomic DNA purification kit (ThermoFisher Scientific, USA, #K0721) was used to extract genomic DNA in accordance with the manufacturer's protocols. *B. pseudomallei* clinical isolates were obtained from 33 cases of melioidosis patients admitted to Sunpasitthiprasong hospital in Ubon Ratchathani, Thailand. Samples came from either blood, urine, or throat swabs of melioidosis patients and cultured on Ashdown's agar to select for the growth of *B. pseudomallei* [103]. For each sample, a single colony was picked and streaked on a Columbia agar and incubated at 37˚C overnight. A loop full of *B. pseudomallei* from the Columbia agar was then inoculated in 3 mL Luria-Bertani broth and incubated at 37˚C with shaking at 200 rpm for 16–18 hours in the biosafety level 3 laboratory. *B. pseudomallei* was harvested and lysed for genomic DNA extraction using QIAmp Mini kit protocol (QIAGEN, Germany, #51304).

## Results

### Designing CRISPR-Cas12a specific to *B. pseudomallei*

CRISPR-Cas12a is an RNA guided-ribonucleoprotein (RNP) complex that recognizes its target DNA through base-pairing complementarity [104–106]. Recognition and cleavage of the target transform the RNP complex into a non-specific nuclease that collaterally cleaves single-stranded DNA (ssDNA) *in vitro* [44,107]. This collateral cleavage property and programmability of CRISPR-Cas12a offer an adaptation for highly specific and sensitive detection of DNA [44,46,53,108–110]. Therefore, we aimed to apply this CRISPR diagnostic platform to the detection of *B. pseudomallei* DNA.

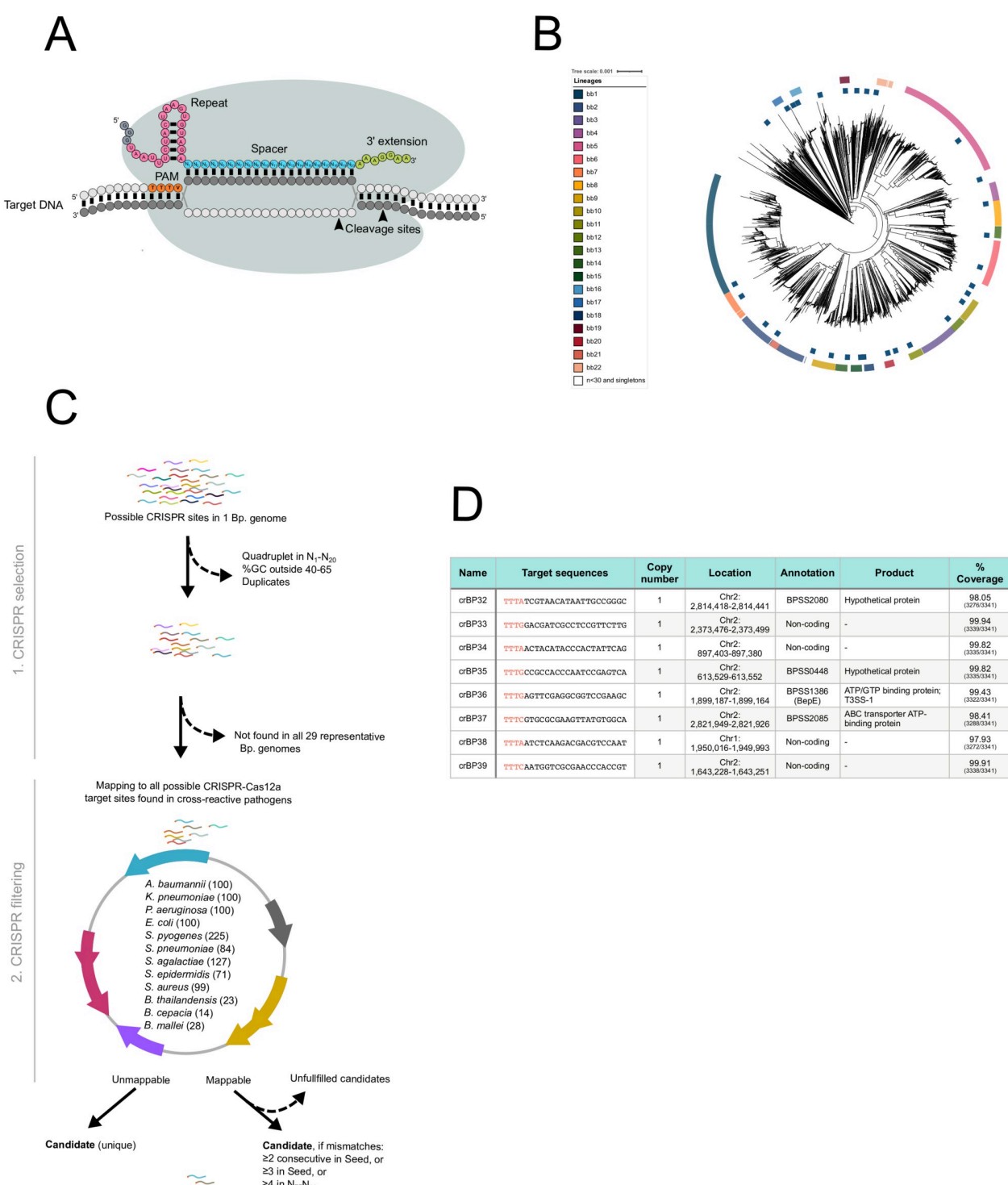

**Fig 1. Designing *B. pseudomallei*-specific crRNA.** (A) A schematic representation of CRISPR-Cas12a recognizing its target DNA. In forming a CRISPR-Cas12a RNP complex, Cas12a protein (grey lobes) needs to interact with crRNA. The crRNA consists of two essential elements—a repeat (pink dots) and a spacer (blue dots). The repeat is a structural scaffold of the crRNA, required for crRNA-Cas12a interaction, while the spacer is used for target recognition. An extra 3' extension (green dots) of the RNA is added in this study (see main text). Recognition of CRISPR-Cas12a RNP complex to its target DNA is specified by base-pairing interaction between a 20-nt spacer of crRNA and a template strand of the target DNA (dark grey dots). PAM (TTTV; orange dots), however, is a prerequisite for this interaction. The target recognition leads to cleavage of the DNA at the PAM-distal sites (black arrow heads) and *trans* cleavage. (B) A phylogeny of global *B. pseudomallei* population (n = 3,341) collected between years of 1935 and 2018 (see Materials and Methods). The outer ring is colored coded by major lineages denoted as the *Burkholderia* Bayesian cluster

(bb) 1 to 22. Blue points located in the inner ring highlight 30 representative genomes used for CRISPR-Cas12a target scanning in Fig 1C. (C) A schematic representation of the bioinformatic analysis used for identifying *B. pseudomallei*-specific CRISPR-Cas12a target sites. The analysis consists of 2 steps—CRISPR selection and CRISPR filtering. In the CRISPR selection, common candidate target sites are generated from 30 representative *B. pseudomallei* genomes found in Fig 1B. These candidates are then filtered through 1,071 genomes of possible cross-reactive pathogens during the CRISPR filtration step. Candidate target sites that are unique or contain significant mismatches are selected as final candidates. (D) A table showing details of eight CRISPR-Cas12a target sites (patent pending) identified from Fig 1C. Target sequences are shown, consisting of a 20-nucleotide spacer, preceded by PAM (red text). Percentage coverage denotes occurrence of each final candidate in the *B. pseudomallei* global genome collection (n = 3,341). Annotation is based on that of the K96243 strain (assembly GCF_000011545.1).

A key to the specificity of CRISPR-Cas12a-based detection is a spacer region of the crRNA [99,100], which base-pairs to target DNA (Fig 1A). Since *B. pseudomallei* is highly recombinogenic with constant reshuffling of genomic contents [71,111–113], a well-designed spacer to target a DNA sequence that is present in all *B. pseudomallei* is needed to ensure high coverage of detection. Thus, we performed a phylogenetic analysis of 3,341 *B. pseudomallei* genomes from a global collection (S1 Table). The analysis classified the *B. pseudomallei* population into 22 major lineages (Fig 1B). We then selected a representative from each major lineage and eight other outsider representatives for our in-house bioinformatic pipeline, which searches for CRISPR-Cas12a candidate target sites (Fig 1C).

In this pipeline, we first generated a pool of optimal target sites that are common among the 30 *B. pseudomallei* representative genomes. These target sites contain TTTV (V is A, C or G) as a protospacer adjacent motif (PAM) on the 5' end followed by a 20-nt long spacer (Fig 1A). The resulting candidate target sites were then filtered through 1,071 genomes of various bacterial pathogens that cause frequent infections or those that are closely related to *B. pseudomallei* including *B. cepacia*, *B. mallei* and *B. thailandensis*. This final step yielded candidate sites that either are unique to *B. pseudomallei* or contain significant mismatches to those cross-reactive pathogens; both of which could be highly specific target sites for CRISPR-Cas12a. The results showed that after executing the bioinformatic pipeline, we retrieved eight elite target sites, each found as a single copy in a *B. pseudomallei* genome (Fig 1D). These target sites are located in both coding and non-coding regions of the genome. Using *in silico* search for potential cross-reactive sites against the NCBI genomic database [114] (data retrieved on May 5th, 2022), we further confirmed a high specificity of all target sites with no cross-reaction with human [101] or other bacterial genomes [102] (S3 and S4 Figs, and Materials and Methods). In contrast, when analyzing coverage of these eight target sites on a *B. pseudomallei* global population, we found that all of them can be found in at least in 97% of the population (Fig 1D).

## Highly specific detection of *B. pseudomallei* DNA by crBP34, 36 and 38

To verify that the identified target sites can be used with CRISPR-Cas12a to detect *B. pseudomallei* genomic DNA, we engineered CRISPR-Cas12a to contain target sites in the spacer region of crRNA and applied these modified CRISPR-Cas12a enzyme to DETECTR assay (Fig 2A). In the DETECTR assay, the pathogen's DNA is pre-amplified by an isothermal method before being combined with pre-assembled CRISPR-Cas12a RNP in a CRISPR reaction [44]. This reaction contains an excess amount of short ssDNA probes modified with fluorescein and quencher on the 5' and 3' ends (FAM-Quencher probe), respectively. In the presence of the pathogen's DNA, a programmed CRISPR-Cas12a RNP complex recognizes an intended target and activates itself into a non-specific ssDNAse that cleaves the probes in *trans*. Liberation of the fluorescein from the quencher results in an emission of fluorescence, a proxy for detection of the pathogen's DNA.

We designed 35-nt primers (S3 Table) and performed RPA on only seven target sites. The exception was crBP33 of which optimal primers could not be designed due to highly repetitive

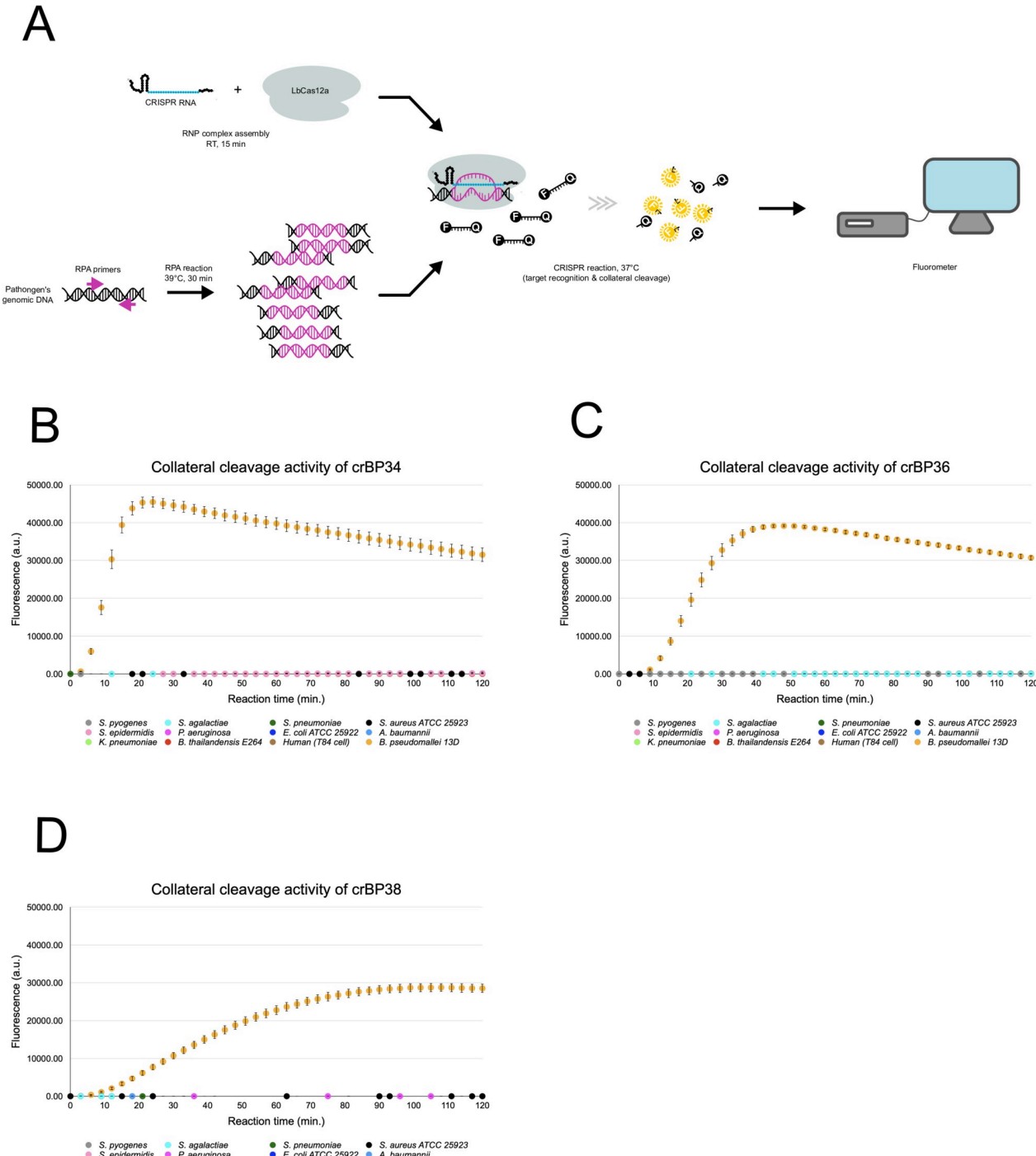

**Fig 2. Specific detection of *B. pseudomallei* genomic DNA by the designed CRISPR-Cas12a.** (A) A schematic representation of DETECTR assay. First, a region containing the CRISPR-Cas12a target site in the pathogen's genomic DNA is pre-amplified by flanking primers (magenta arrows) in an isothermal amplification reaction called RPA. This amplified DNA is then combined with the pre-assembled CRISPR-Cas12a RNP complex in a CRISPR reaction that contains FAM-Quencher ssDNA probes (F-Q). Recognition of CRISPR RNP to its target self-activates its *trans* cleavage (collateral cleavage) activity. This results in cleavage of the FAM fluorophore from its quencher, hence the generation of a fluorescent signal, which can be detected by fluorometer. (B, C, D) Collateral cleavage activity of crBP34, crBP36 and crBP38, respectively. Each CRISPR-Cas12a RNP complex was tested against genomic DNA from a human cell line and 11 other pathogens, including *B. thailandensis* and *B. psedomallei*. ~10,000 copies of genomic DNA were used in 40-μl RPA reactions, of which 3 μl were used in 100-μl CRISPR reactions. Fluorescent signals were collected every three minutes. Signals from CRISPR reactions that contained water-input RPA were used for background subtraction. Each data point represents an average from six CRISPR reactions derived from two independent sets of RPA reactions. Note that some data points sporadically

become slightly negative after background subtraction, causing them to be invisible on the scale. Standard deviation is also plotted. a.u.: arbitrary units.

sequences and GC-rich nature in this region. The result showed that RPA of target sites crBP34, crBP36 and crBP38 yielded good amplification products among the seven target sites (S1 Fig). Therefore, we selected these candidate target sites and engineered their 20-nt sequences (excluding PAM) into the spacer region of the crRNA (Fig 1A). The 7-nt RNA extension was also added to the 3' end of the crRNA, as it has been shown to increase collateral cleavage activity [115]. Genomic DNA from 11 pathogens that are common causes of bacterial infections in melioidosis endemic areas, including *S. pyogenes*, *S. epidermidis*, *K. pneumoniae*, *S. agalactiae*, *P. aeruginosa*, *B. thailandensis*, *S. pneumoniae*, *E. coli*, *S. aureus*, *A. baumannii* and *B. pseudomallei* were pre-amplified using RPA and tested with each crRNA. The results of fluorescence activation showed that all three crRNAs could specifically detect *B. pseudomallei*, but not other pathogens (Fig 2B, 2C and 2D). Encouragingly, all three crRNAs could discriminate effectively against *B. pseudomallei*'s close relative *B. thailandensis*. This specificity exceeded our expectation since fluorescent signals from non-*B. pseudomallei* bacteria were barely above the background level, in contrast to that from *B. pseudomallei* which exhibited 4-order of magnitude of intensity. Since there is the potential to use this platform to diagnose melioidosis in humans, we also tested crBP34, 36 and 38 with human genomic DNA. We found that all three crRNAs did not cross-react with human DNA (Fig 2B, 2C and 2D). These results validated our crRNA design pipeline that excluded cross-reaction from non-*B. pseudomallei* DNA (Fig 1A). In addition, we noticed that each crRNA exhibited differing kinetics of detection. crBP34 had the strongest signal that reached a maximum at 24 minutes, while crBP36 reached a maximum at 48 minutes. crBP38, on the other hand, displayed the most delayed maximum at 108 minutes. These variations could be a result of cumulative differences in both RPA efficiency and the intrinsic property of each crRNA. Regardless, all three crRNAs displayed specific signals above the background in less than 10 minutes into the reaction. Since crBP34 possesses the strongest signal and the fastest kinetics, we selected crBP34 for the subsequent experiments.

## Sensitive and rapid detection of *B. pseudomallei* DNA by crBP34

One important feature of a reliable diagnostic tool is sensitivity. To investigate the ability of crBP34-coupled DETECTR in detecting trace amounts of genetic material, frequently present in clinical samples, we determined the limit of detection (LOD) of this CRISPR platform. A known copy number of genomic DNA of *B. pseudomallei* was serially diluted and used in the assay as described above. The result showed that our CRISPR-Cas12a detection platform has an LOD at 40 copies of *B. pseudomallei* genomic DNA per reaction (Fig 3A). In fact, the assay could detect as low as four copies of input genomic DNA; however, at this lower DNA amount, detection was inconsistent, and the fluorescent signal was weaker and had a more delayed rate of detection (maximum at 55 minutes). For comparison, we also performed real-time PCR—a sensitive standardized method for nucleic acid detection. When comparing the LOD of crBP34-DETECTR to that of real-time PCR, we found that our detection platform performed equally well as did this standard method, which showed reliable detection at 40 copies per reaction (Fig 3B). In addition, we tested this detection platform with *B. pseudomallei* genomic DNA spiked with different amounts of human genomic DNA. As expected, the result showed that the sensitivity was negatively impacted by the presence of human DNA background, especially at LOD of the assay (S2 Fig).

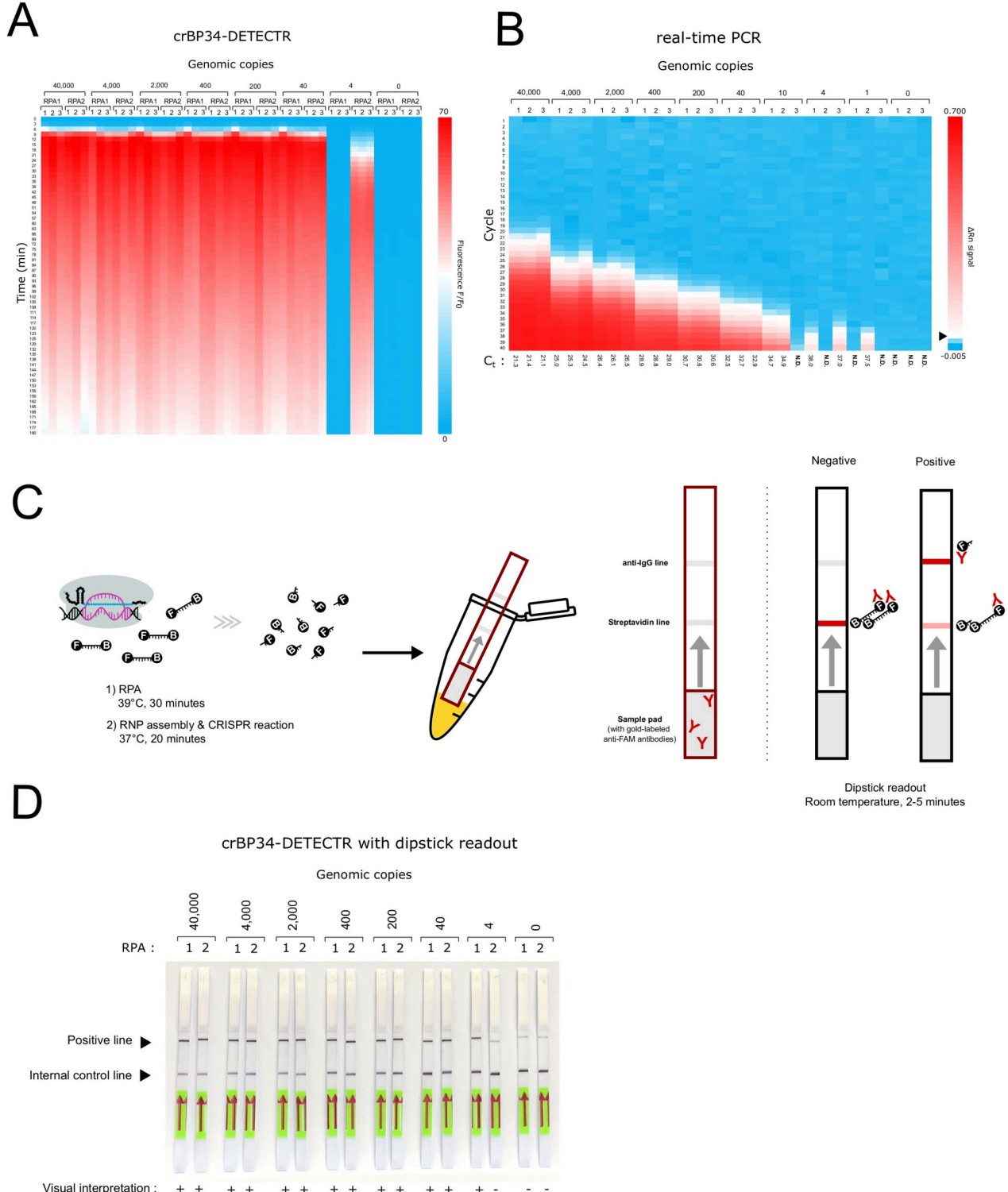

**Fig 3. crBP34-DETECTR shows comparable sensitivity to real-time PCR.** (A) A heatmap showing fluorescent signals generated by crBP34-DETECTR assay. Known copy numbers of *B. pseudomallei* genomic DNA ranging from 40,000–0 copies were used in 50-μl RPA reactions, of which 15 μl were used in 100-μl CRISPR reactions. Fluorescent signals were collected at three-minute intervals. The heatmap denotes fluorescence at indicated times compared to fluorescence at 0 minutes ($F/F_0$). For each copy number, two independent RPA reactions were performed; each was subjected to three replicates of CRISPR reactions (total of six CRISPR reactions). A scale bar is shown on the right. (B) A heatmap showing fluorescent signals generated by real-time PCR assay. Known copy numbers of *B. pseudomallei* genomic DNA ranging from 40,000–0 copies were used in 20-μl PCR reactions. The heatmap denotes fluorescent signals (ΔRn) as a function of the PCR cycle. Datasets from three replicates are

presented. A scale bar is shown on the right. A black arrowhead indicates a threshold value that is used for calculating threshold cycles ($C_t$), presented under the heatmap. (C) A schematic representation of a lateral flow dipstick readout. RPA and CRISPR reactions were performed similar to Fig 2A, except that FAM-Quencher probes are replaced with FAM-Biotin probes (F-B). Note that RNP assembly is done simultaneously in the CRISPR reaction. Following this, a sample pad of a dipstick is directly immerged into the CRISPR reaction to allow labeling of the probes by anti-FAM antibodies and diffusion into the dipstick. Positive detection is indicated by the presence of an intensified band at the anti-IgG line. The streptavidin line serves as an internal control (see main text). (D) Lateral flow dipstick readout retains sensitivity of the crBP34-DETECTR assay. Known copy numbers of *B. pseudomallei* genomic DNA ranging from 40,000–0 copies were used in 30-μl RPA reactions, of which 5 μl were used in 50-μl CRISPR reactions that contained FAM-Biotin probes. Dipsticks were used as a readout of the assay, as explained in c). Two sets of RPA reactions were performed. Visual interpretation was done by comparing the intensity of a positive band to its background at 0 copy genomic DNA.

To obviate the requirement of an expensive laboratory setup, we replaced detection of fluorescence with a lateral flow dipstick for the assay readout. Consequently, the FAM-Quencher ssDNA probes were substituted by FAM-Biotin ssDNA probes in the CRISPR reaction. Afterwards, a sample pad of a lateral flow dipstick was directly immersed into the CRISPR reaction. This sample pad contains gold-conjugated anti-FAM antibodies, which label FAM moieties for visualization—both in intact probes and in cleaved probes (Fig 3C). In the absence of *B. pseudomallei* target DNA, the probes are intact and trapped at the streptavidin line by biotin-streptavidin interaction. However, in the presence of the target DNA, collateral activity of CRISPR-Cas12a degrades the probes, liberating the FAM-antibody complexes from biotin so that they can diffuse further on the dipstick to bind to the anti-IgG line. We tested this readout with various amounts of *B. pseudomallei* genomic DNA and found that the sensitivity of this readout correlates well with that of the fluorescence readout, with an LOD at 40 copies per reaction (Fig 3D). Therefore, the lateral flow dipstick can be used to give a sensitive readout of the assay, providing an easy method for detection of *B. pseudomallei*.

## crBP34 unambiguously detects *B. pseudomallei* DNA from clinical isolates

*In silico* analysis indicated that all eight identified CRISPR-Cas12a target sites were highly conserved. crBP34 has sequences that match 3,335 out of 3,341 *B. pseudomallei* genomes (Fig 1D) and were shown to be very specific in our preliminary data (Fig 2B). To verify that our designed crBP34 can specifically detect other *B. pseudomallei* isolates, we tested our CRISPR diagnostic platform with various clinically isolated bacteria, specifically *B. pseudomallei* and common non-*B. pseudomallei* pathogens. The results showed that crBP34 could accurately detect all *B. pseudomallei* isolates (n = 33) (Fig 4A and 4B). Importantly, the assay discriminated against all other DNA from the non-*B. pseudomallei* pathogens (n = 88). The fluorescence signals from this latter group of pathogens remained at a baseline, at the same level of the background (water-input RPA, 'None'). Together, these data suggest that crBP34 exhibits 100% specificity and could potentially be used to detect *B. pseudomallei* from different geographical areas.

## Discussion

Nucleic acid detection by isothermal amplification assays such as RPA and LAMP have had increased usages in research communities due to their high sensitivity and simplicity. Nevertheless, LAMP demands a series of six to eight primers to work together in an optimal locus [116]. This requirement could possibly limit its broad usage in the detection of *B. pseudomallei*, which is heterogenous and GC-rich in its genome. RPA, on the other hand, can tolerate significant mismatches in primers [32], making it less specific. We often observed non-specific amplification in our RPA reactions. In comparison, the DETECTR assay combines RPA and CRISPR-Cas12a. It utilizes CRISPR-Cas12a as a sequence-specific DNA reader to verify the identity of pre-amplified DNA from RPA reaction. This adds an extra stringency to RPA while

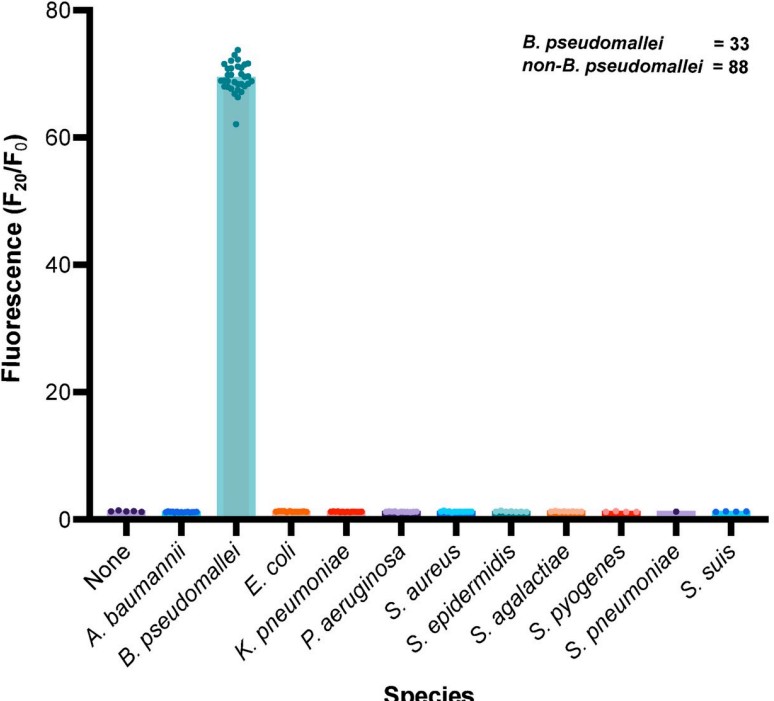

**A**

### Detection of clinical isolates with crBP34-DETECTR

*B. pseudomallei* = 33
non-*B. pseudomallei* = 88

**B**

### Descriptive statistics

| Species | Number of values | $F_{20}/F_0$ | | | | |
|---|---|---|---|---|---|---|
| | | Minimum | Maximum | Range | Mean | Std. Deviation |
| None | 5 | 1.20 | 1.44 | 0.25 | 1.31 | 0.09 |
| *A. baumannii* | 15 | 1.10 | 1.27 | 0.17 | 1.17 | 0.05 |
| ***B. pseudomallei*** | 33 | 62.09 | 73.73 | 11.64 | 69.37 | 2.25 |
| *E. coli* | 12 | 1.16 | 1.32 | 0.17 | 1.25 | 0.06 |
| *K. pneumoniae* | 14 | 1.15 | 1.28 | 0.14 | 1.22 | 0.04 |
| *P. aeruginosa* | 11 | 1.12 | 1.29 | 0.17 | 1.21 | 0.05 |
| *S. aureus* | 12 | 1.13 | 1.40 | 0.27 | 1.24 | 0.06 |
| *S. epidermidis* | 7 | 1.20 | 1.38 | 0.18 | 1.25 | 0.06 |
| *S. agalactiae* | 8 | 1.24 | 1.36 | 0.12 | 1.26 | 0.04 |
| *S. pyogenes* | 4 | 1.19 | 1.31 | 0.12 | 1.23 | 0.05 |
| *S. pneumoniae* | 1 | 1.24 | 1.24 | 0.00 | 1.24 | 0.00 |
| *S. suis* | 4 | 1.20 | 1.30 | 0.10 | 1.25 | 0.04 |

**Fig 4. crBP34 specifically detects clinically isolated *B. pseudomallei*.** (A) A graph displaying fluorescent activation ($F_{20}/F_0$) values from crBP34-DETECTR assay. 1 ng of genomic DNA from each bacterial isolate was used in 30-μl RPA reactions, of which 5 μl were used in 100-μl CRISPR reactions. Fluorescent signals were collected at time points 0 and 20 minutes and used for calculating $F_{20}/F_0$ values. Each dot represents a single replicate from each isolate. (B) A table detailing statistical data of Fig 4A.

maintaining its high sensitivity. As a result, our crBP34-DETECTR assay is highly specific and sensitive. The assay has an LOD at 40 copies per reaction, comparable to that of real-time PCR (Fig 3B), LAMP [26] and RPA [29,30].

The specificity of the crBP34-DETECTR may arise from the designed crRNA—crBP34, which was a result of large-scale phylogenetics and bioinformatics analyses to remove cross-reactivity (Fig 1B and 1C). This contrasts with previously reported detection assays using real-time PCR [21–25], LAMP [26,27] or RPA [28–30], in which primers were designed based on selected genomic loci or limited bioinformatics analysis on few reference strains. Testing crBP34 and the other crRNAs with *B. pseudomallei* isolates from other geographical areas remains for further examination to assess their universal application. However, the coverage analysis predicted that both the CRISPR-Cas12 target sites (Fig 1D) and RPA primers (S4 Table) are present in more than 97% of the global *B. pseudomallei* population. This gives a high probability of detection by our designed crRNAs.

Even though we are reporting here that our stand-alone crBP34-DETECTR performed exceptionally well in the detection of the DNA from clinically isolated *B. pseudomallei*, direct application of this platform to clinical specimens such as sputum, blood or urine is not presented in this study. Not only is DNA extraction from these samples laborious and challenging, but it also does yield an overwhelming amount of human DNA that negatively impacts sensitivity and specificity of subsequent nucleic acid detections [117–121]. When mixing human DNA with *B. pseudomallei* DNA, we observed a reduction of the RPA product, resulting in a subsequent loss of collateral activity of the CRISPR-Cas12a (S2 Fig). Hence, DNA extraction methodology that selectively enriches pathogens from clinical samples will play a crucial role in the detection. Although commercial kits are available, the added extra steps and high cost for diagnosis have prevented their usage in facilities with limited resources. Therefore, development of a streamlined method for processing bacterial genomic DNA from clinical specimens is much needed, especially one that is compatible with the DETECTR assay; this is a subject of our future study.

In conclusion, we have developed a novel method for the detection of *B. pseudomallei* genomic DNA using CRIPSR-Cas12a. Our detection platform, crBP34-DETECTR, is highly specific for *B. pseudomallei* as it detected all clinical isolates of endemic *B. pseudomallei*, while discriminating against human and other pathogens including its closely related species *B. thailandensis* (Fig 2B, 4A and 4B). The adaptation of a lateral flow dipstick for a readout enabled this assay to be performed in a regular laboratory, without the requirement of specific instruments and the loss of sensitivity (Fig 3C and 3D). This crBP34-DETECTR platform will facilitate point-of-care or field-deployable diagnoses in the future.

## Supporting information

**S1 Fig. RPA product of each target site.**
(TIF)

**S2 Fig. Effects of human genomic DNA on RPA reaction.**
(TIF)

**S3 Fig. Targets of crBP sequence in other bacterial species.**
(TIF)

**S4 Fig. Targets of crBP sequence in the human genome.**
(TIF)

**S1 Table. Global *B. pseudomallei* genome collection.**
(XLSX)

**S2 Table. 30 *B. pseudomallei* representatives.**
(XLSX)

**S3 Table. DNA oligonucleotides and probes.**
(XLSX)

**S4 Table. Coverage of RPA primers in a global *B. pseudomallei* genome collection.**
(XLSX)

**S5 Table. 1,071 complete genome assemblies of Non-*B. pseudomallei* bacteria.**
(XLSX)

## Acknowledgments

The authors thank the Erawan HPC Project of the Information Technology Service Center (ITSC), Chiang Mai University, Chiang Mai, Thailand for providing computational resources. In addition, we thank Dr. Chayasith Uttamapinant (VISTEC, Thailand) for providing useful suggestions and reagents, and Dr. Direk Limmathurotsakul (MORU, Thailand) for a helpful discussion.

## Author Contributions

**Conceptualization:** Somsakul Pop Wongpalee, Hathairat Thananchai, Claire Chewapreecha.

**Formal analysis:** Somsakul Pop Wongpalee, Claire Chewapreecha, Henrik B. Roslund, Chalita Chomkatekaew.

**Funding acquisition:** Somsakul Pop Wongpalee.

**Investigation:** Somsakul Pop Wongpalee, Warunya Tananupak, Phumrapee Boonklang, Sukritpong Pakdeerat.

**Resources:** Claire Chewapreecha, Phumrapee Boonklang, Sukritpong Pakdeerat, Rathanin Seng, Narisara Chantratita, Piyawan Takarn, Phadungkiat Khamnoi.

**Software:** Claire Chewapreecha, Henrik B. Roslund, Chalita Chomkatekaew.

**Supervision:** Somsakul Pop Wongpalee.

**Writing – original draft:** Hathairat Thananchai.

**Writing – review & editing:** Somsakul Pop Wongpalee, Claire Chewapreecha.

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
