## [Decision Letter · Decision Letter 0]

26 Apr 2022

Dear Dr. Wongpalee,

Thank you very much for submitting your manuscript "Highly Specific and Sensitive Detection of Burkholderia pseudomallei Genomic DNA by CRISPR-Cas12a" for consideration at PLOS Neglected Tropical Diseases. As with all papers reviewed by the journal, your manuscript was reviewed by members of the editorial board and by several independent reviewers. In light of the reviews (below this email), we would like to invite the resubmission of a significantly-revised version that takes into account the reviewers' comments. 

We cannot make any decision about publication until we have seen the revised manuscript and your response to the reviewers' comments. Your revised manuscript is also likely to be sent to reviewers for further evaluation.

Sincerely,

Alfredo G Torres

Deputy Editor

Alfredo Torres

Deputy Editor

Reviewer's Responses to Questions

**Key Review Criteria Required for Acceptance?**

**Methods**

-Are the objectives of the study clearly articulated with a clear testable hypothesis stated?

-Is the study design appropriate to address the stated objectives?

-Is the population clearly described and appropriate for the hypothesis being tested?

-Is the sample size sufficient to ensure adequate power to address the hypothesis being tested?

-Were correct statistical analysis used to support conclusions?

-Are there concerns about ethical or regulatory requirements being met?

Reviewer #1: The manuscript entitled with " Highly Specific and Sensitive Detection of Burkholderia pseudomallei Genomic DNA by CRISPR-Cas12a" submitted by Wongpalee et.al., presents a novel method that circumvents long time amplification issues by utilizing CRISPR-Cas12a coupled with isothermal amplification of a target DNA. Although the article has scientific rigor, few flaws need to be corrected before it can be considered for publication.

• Aspects of CRISPR technology for diagnostics and treatment needs more advanced elaboration.

https://link.springer.com/article/10.1007/s40097-022-00472-7

• Define the “frequency” and the time duration for sonication in the method protocol.

• Report the length, melting temperature and probes of the primers used in the RPA and the region used for the amplification.

• Correct the “µl” spelling throughout the document

• Define the importance of the target sites “crBP34”, “crBP36”, “crBP38” and its role as a controlling factor for melioidosis.

• Are the researchers making any modifications in the purification protocol, as there is a lack of component “sodium phosphate” in both the binding and eluting buffer. The researchers are requested to share the purification profile 

• “The results showed that all three crRNAs could specifically detect B. pseudomallei, but not other pathogens”. Are the authors taking any range level to specify the detection limit, as it is observed form the Figure 2B: the fluorescence intensity is detected for the “S. epidermidis” and “B. thailandensis” as well apart from “B. pseudomallei”; and in Figure 2C: the fluorescence intensity is detected for the “S. pneumoniae” and “S. aureus ATCC” also on a lower scale.

Reviewer #2: The authors had robust methods that were thoroughly explained. My concerns/questions regarding the methods are listed below.

Line 136: Define LB. Was Lennox broth or Luria Bertani broth used? Line 290 defines LB as Luria-Bertani but it would be useful to have LB defined on line 136 as well.

It was unclear to me how the authors selected crBP34, 36 and 38 from the list of candidate target sites from Fig 1D. 

Fig 1 was a very helpful figure to understand the design process for the Bp specific CRISPR RNA. I was happy to see that the authors screened the possible CRISPR sites against cross-reactive pathogens and I have two questions about this step. 1) Why did the authors not in-silico screen the final target (crBP34) against all available bacterial pathogens from NCBI. 2) If the ultimate goal of this assay is to test clinical samples why were the targets not screened against the human genome to look for interactions?

**Results**

-Does the analysis presented match the analysis plan?

-Are the results clearly and completely presented?

-Are the figures (Tables, Images) of sufficient quality for clarity?

Reviewer #1: The manuscript entitled with " Highly Specific and Sensitive Detection of Burkholderia pseudomallei Genomic DNA by CRISPR-Cas12a" submitted by Wongpalee et.al., presents a novel method that circumvents long time amplification issues by utilizing CRISPR-Cas12a coupled with isothermal amplification of a target DNA. Although the article has scientific rigor, few flaws need to be corrected before it can be considered for publication.

• Aspects of CRISPR technology for diagnostics and treatment needs more advanced elaboration.

https://link.springer.com/article/10.1007/s40097-022-00472-7

• Define the “frequency” and the time duration for sonication in the method protocol.

• Report the length, melting temperature and probes of the primers used in the RPA and the region used for the amplification.

• Correct the “µl” spelling throughout the document

• Define the importance of the target sites “crBP34”, “crBP36”, “crBP38” and its role as a controlling factor for melioidosis.

• Are the researchers making any modifications in the purification protocol, as there is a lack of component “sodium phosphate” in both the binding and eluting buffer. The researchers are requested to share the purification profile 

• “The results showed that all three crRNAs could specifically detect B. pseudomallei, but not other pathogens”. Are the authors taking any range level to specify the detection limit, as it is observed form the Figure 2B: the fluorescence intensity is detected for the “S. epidermidis” and “B. thailandensis” as well apart from “B. pseudomallei”; and in Figure 2C: the fluorescence intensity is detected for the “S. pneumoniae” and “S. aureus ATCC” also on a lower scale.

Reviewer #2: The results are described and presented in a clear manner. The figures are detailed and provide a nice visual of the design and validation process.

**Conclusions**

-Are the conclusions supported by the data presented?

-Are the limitations of analysis clearly described?

-Do the authors discuss how these data can be helpful to advance our understanding of the topic under study?

-Is public health relevance addressed?

Reviewer #1: The manuscript entitled with " Highly Specific and Sensitive Detection of Burkholderia pseudomallei Genomic DNA by CRISPR-Cas12a" submitted by Wongpalee et.al., presents a novel method that circumvents long time amplification issues by utilizing CRISPR-Cas12a coupled with isothermal amplification of a target DNA. Although the article has scientific rigor, few flaws need to be corrected before it can be considered for publication.

• Aspects of CRISPR technology for diagnostics and treatment needs more advanced elaboration.

https://link.springer.com/article/10.1007/s40097-022-00472-7

• Define the “frequency” and the time duration for sonication in the method protocol.

• Report the length, melting temperature and probes of the primers used in the RPA and the region used for the amplification.

• Correct the “µl” spelling throughout the document

• Define the importance of the target sites “crBP34”, “crBP36”, “crBP38” and its role as a controlling factor for melioidosis.

• Are the researchers making any modifications in the purification protocol, as there is a lack of component “sodium phosphate” in both the binding and eluting buffer. The researchers are requested to share the purification profile 

• “The results showed that all three crRNAs could specifically detect B. pseudomallei, but not other pathogens”. Are the authors taking any range level to specify the detection limit, as it is observed form the Figure 2B: the fluorescence intensity is detected for the “S. epidermidis” and “B. thailandensis” as well apart from “B. pseudomallei”; and in Figure 2C: the fluorescence intensity is detected for the “S. pneumoniae” and “S. aureus ATCC” also on a lower scale.

Reviewer #2: The discussion is concise and describes the power of the newly developed crBP34-DETECTR in both sensitivity and specificity. The authors also reiterated that the assay is currently validated for the detection of Bp DNA after isolation from a clinical sample and more work is needed to validate the assay on complex clinical samples.

**Editorial and Data Presentation Modifications?**

Reviewer #1: The manuscript entitled with " Highly Specific and Sensitive Detection of Burkholderia pseudomallei Genomic DNA by CRISPR-Cas12a" submitted by Wongpalee et.al., presents a novel method that circumvents long time amplification issues by utilizing CRISPR-Cas12a coupled with isothermal amplification of a target DNA. Although the article has scientific rigor, few flaws need to be corrected before it can be considered for publication.

• Aspects of CRISPR technology for diagnostics and treatment needs more advanced elaboration.

https://link.springer.com/article/10.1007/s40097-022-00472-7

• Define the “frequency” and the time duration for sonication in the method protocol.

• Report the length, melting temperature and probes of the primers used in the RPA and the region used for the amplification.

• Correct the “µl” spelling throughout the document

• Define the importance of the target sites “crBP34”, “crBP36”, “crBP38” and its role as a controlling factor for melioidosis.

• Are the researchers making any modifications in the purification protocol, as there is a lack of component “sodium phosphate” in both the binding and eluting buffer. The researchers are requested to share the purification profile 

• “The results showed that all three crRNAs could specifically detect B. pseudomallei, but not other pathogens”. Are the authors taking any range level to specify the detection limit, as it is observed form the Figure 2B: the fluorescence intensity is detected for the “S. epidermidis” and “B. thailandensis” as well apart from “B. pseudomallei”; and in Figure 2C: the fluorescence intensity is detected for the “S. pneumoniae” and “S. aureus ATCC” also on a lower scale.

Reviewer #2: Minor:

Line 105, uncapitalize “Initial”

Line 270: uncapitalize “Primers”

Line 288: It looks like “incubate” should be in the past tense “incubated”

Line 465: “figure 2a” instead of Fig 2a

**Summary and General Comments**

Reviewer #1: The manuscript entitled with " Highly Specific and Sensitive Detection of Burkholderia pseudomallei Genomic DNA by CRISPR-Cas12a" submitted by Wongpalee et.al., presents a novel method that circumvents long time amplification issues by utilizing CRISPR-Cas12a coupled with isothermal amplification of a target DNA. Although the article has scientific rigor, few flaws need to be corrected before it can be considered for publication.

• Aspects of CRISPR technology for diagnostics and treatment needs more advanced elaboration.

https://link.springer.com/article/10.1007/s40097-022-00472-7

• Define the “frequency” and the time duration for sonication in the method protocol.

• Report the length, melting temperature and probes of the primers used in the RPA and the region used for the amplification.

• Correct the “µl” spelling throughout the document

• Define the importance of the target sites “crBP34”, “crBP36”, “crBP38” and its role as a controlling factor for melioidosis.

• Are the researchers making any modifications in the purification protocol, as there is a lack of component “sodium phosphate” in both the binding and eluting buffer. The researchers are requested to share the purification profile 

• “The results showed that all three crRNAs could specifically detect B. pseudomallei, but not other pathogens”. Are the authors taking any range level to specify the detection limit, as it is observed form the Figure 2B: the fluorescence intensity is detected for the “S. epidermidis” and “B. thailandensis” as well apart from “B. pseudomallei”; and in Figure 2C: the fluorescence intensity is detected for the “S. pneumoniae” and “S. aureus ATCC” also on a lower scale.

Reviewer #2: In the introduction the authors set up the issue well and why they aimed to find a cheaper, specific, fast alternative to detect Bp in clinical samples. The authors reviewed the literature and although there have been other studies that have developed a lateral flow recombinase polymerase amplification (LF-RPA) for the detection of Bp the authors of this study point out that the previously published studies did not investigate the specificity of the targets were not tested on a wider Bp population. Additionally, this study is unique from the other studies since it also uses CRISPR-Cas12a for the detection of Bp which has been validated on other pathogen species. 

This study seems like a big step in the right direction for a cheaper, faster alternative for Bp detection in the clinic. The only major limitation I see with this study is the lack of validating the assay in the presence of a human DNA background. The authors speak to the limitation of the study in that clinically isolated Bp and not clinical samples (blood, urine, etc.) were not tested (lines 529-530) but they don’t speak of the potential interactions that may occur with the human DNA background present in most clinical samples. With this said it should be noted that the authors do not make claims that their assay detects anything more than Bp DNA. The title is quite clear that this assay is detecting Bp genomic DNA.

The abstract is a bit misleading since the authors say that the assay can be achieved in less than 1 hour (line 34) but fail to mention that this is after isolating Bp from a clinical sample (24-48 hours) and after a DNA extraction on that pure isolate (1-4 hours). 

I think the authors need to be clearer about the input material required for the assay. For example, lines 36-37 make claims of this assay being used in “poor-resourced clinical settings” which may be true one day but that is not the case yet since the assay has not been validated on clinical samples, DNA extractions from clinical samples, or mock clinical DNA samples of Bp DNA with a human DNA background. Where lines 527-530 in the discussion are clearer about this limitation. I would like to see something more like lines 527-530 in the abstract instead of lines 36-37.

PLOS authors have the option to publish the peer review history of their article (what does this mean?). If published, this will include your full peer review and any attached files.

Reviewer #1: No

Reviewer #2: No
---

## [Decision Letter · Decision Letter 1]

12 Jul 2022

Dear Dr. Wongpalee,

We are pleased to inform you that your manuscript 'Highly Specific and Sensitive Detection of Burkholderia pseudomallei Genomic DNA by CRISPR-Cas12a' has been provisionally accepted for publication in PLOS Neglected Tropical Diseases.

Best regards,

Alfredo G Torres

Section Editor

Alfredo Torres

Section Editor

Reviewer's Responses to Questions

**Key Review Criteria Required for Acceptance?**

**Methods**

-Are the objectives of the study clearly articulated with a clear testable hypothesis stated?

-Is the study design appropriate to address the stated objectives?

-Is the population clearly described and appropriate for the hypothesis being tested?

-Is the sample size sufficient to ensure adequate power to address the hypothesis being tested?

-Were correct statistical analysis used to support conclusions?

-Are there concerns about ethical or regulatory requirements being met?

Reviewer #2: Yes

Reviewer #3: Yes

Reviewer #4: The objectives are clearly highlighted in the manuscript.

**Results**

-Does the analysis presented match the analysis plan?

-Are the results clearly and completely presented?

-Are the figures (Tables, Images) of sufficient quality for clarity?

Reviewer #2: Yes

Reviewer #3: Yes

Reviewer #4: Yes, the results are sufficient, clearly presented.

**Conclusions**

-Are the conclusions supported by the data presented?

-Are the limitations of analysis clearly described?

-Do the authors discuss how these data can be helpful to advance our understanding of the topic under study?

-Is public health relevance addressed?

Reviewer #2: Yes

Reviewer #3: Yes

Reviewer #4: Please improve the conclusion:

1. Please add the recombinase polymerase amplification in the conclusion.

2. Please summarise the comparison between the RPA-CRISPR Cas12a-LFA

3. Please add the heating block or water bath for RPA

**Editorial and Data Presentation Modifications?**

Reviewer #2: Accept

Reviewer #3: (No Response)

Reviewer #4: Suggestion: to change the title to ‘highly specific and sensitive lateral flow assay based-detection of Burkholderia pseudomallei via recombinase polymerase (RPA)-CRISPR-Cas12a-Lateral flow assay

**Summary and General Comments**

Reviewer #2: The authors have addressed all of my previous comments and concerns.

Reviewer #3: I very much enjoyed reviewing this article. The authors should be commended for the thoroughness of experimentation with regards to sensitivity and specificity. The clear explanation of all methodology is refreshing. Having additionally read the initial reviewer comments, I believe this revision is excellent. I look forward to the follow-up experimentation direct from clinical specimen, which will be of even greater use.

Major comments:

Line 79 – 80: a positive serological assay in a patient with prior seroconversion may not be a false positive. Rather, this would may be a true positive that represents past exposure as opposed to active infection.

Line 288: first mention of the DETECTR which is the method created by Chen J et al. It appears ambiguous from the text that this method was created by the authors. Would suggest a simple clarification in the introduction.

Minor comments:

The first use of abbreviations throughout this manuscript should be reassessed. There are several instances where first use of abbreviation was not accompanied with the full description. Examples can be found in line 146, 151, 152, 157, 182.

There are multiple occasions where the instrument or reagent manufacturer is incompletely referenced. Please include the company name and location. Examples include line 154, 158, 269.

Line 248: To enable

Reviewer #4: The author should emphasis on the development of a diagnostic method by utilisation of recombinase polymerase amplification (RPA), CRISPR-Cas12a and lateral flow assay.

PLOS authors have the option to publish the peer review history of their article (what does this mean?). If published, this will include your full peer review and any attached files.

Reviewer #2: **Yes: **Carina M Hall

Reviewer #3: No

Reviewer #4: **Yes: **Ismail Aziah

---

## [Editor Report · Acceptance letter]

12 Aug 2022

Dear Dr. Wongpalee,

We are delighted to inform you that your manuscript, "Highly Specific and Sensitive Detection of Burkholderia pseudomallei Genomic DNA by CRISPR-Cas12a," has been formally accepted for publication in PLOS Neglected Tropical Diseases.

Best regards,

Shaden Kamhawi

co-Editor-in-Chief

Paul Brindley

co-Editor-in-Chief
